# A fossil assemblage from the mid–late Maastrichtian of Gavdos Island, Greece, provides insights into the pre-extinction pelagic ichthyofaunas of the Tethys

Thodoris Argyriou[1,¤a,¤b]*, Apostolos Alexopoulos[2], Jorge D. Carrillo-Briceño[3], Lionel Cavin[4]

1 UMR 7207 (MNHN–Sorbonne Université–CNRS) Centre de Recherche en Paléontologie—Paris, Muséum national d'Histoire naturelle, Paris, France, 2 Faculty of Geology & Geoenvironment, Department of Dynamic, Tectonic & Applied Geology, University of Athens, Athens, Greece, 3 Palaeontological Institute and Museum, University of Zurich, Zurich, Switzerland, 4 Department of Geology and Palaeontology, Natural History Museum, Geneva, Switzerland

¤a Current address: Department of Earth and Environmental Sciences, Paleontology & Geobiology, Ludwig-Maximilians-Universität München, München, Germany
¤b Current address: GeoBio-Center, Ludwig-Maximilians-Universität München, München, Germany
* t.argyriou@lrz.uni-muenchen.de

**Data Availability Statement:** All relevant data are within the paper and its Supporting Information files.

## Abstract

The global body-fossil record of marine 'fishes' from the time interval immediately preceding the Cretaceous–Paleogene Extinction is markedly poor. This deficiency appears to be further exacerbated with regards to offshore and deep-water taxa, obscuring our understanding of the state and composition of corresponding vertebrate faunas at the onset of this major extinction event. Recent fieldwork in the mid–late Maastrichtian exposures of the Pindos Unit in Gavdos Island, Greece, yielded a small but informative sample of fossil 'fishes', which inhabited the Tethys approximately three to four million years before the extinction. In this work we describe this sample, which comprises between eight and nine discrete morphotypes of various size classes, belonging to †Ichthyodectoidei, Aulopiformes (†Dercetidae, †Enchodontidae, †Ichthyotringidae), cf. †Sardinioididae, as well as the hexanchid shark †*Gladioserratus* sp. The new material expands the faunal list for the Maastrichtian of Gavdos Island, and the Pindos Unit as a whole, and further allows for the description of a new genus and species of †Enchodontidae and a new species of †Ichthyotringidae. The two new taxa are found to be widespread in the Maastrichtian of the Pindos Unit. The overall character of the assemblage agrees with previous interpretations of an offshore and rather deep depositional environment for the fossiliferous horizons. Furthermore, it exhibits a higher diversity than, and little taxonomic overlap with penecontemporaneous teleost assemblages from the Tethys, and informs on the otherwise poorly known Maastrichtian offshore and deep-water marine ichthyofaunas of the region.

**Funding:** The collection and study of this material was made possible through the SNSF grant P2ZHP3_184216 to TA. TA was supported by a postdoctoral fellowship of the Alexander von Humboldt Foundation during the final stages of this work. JDCB was supported by SNSF grant 31003A-149605 to Marcelo Sánchez-Villagra.

**Competing interests:** The authors have declared that no competing interests exist.

## Introduction

At the end of the Mesozoic (~66 Ma) global marine ecosystems were faced with the catastrophic effects of Cretaceous–Paleogene (K–Pg) Mass Extinction, which resulted in sharp drops in biodiversity and the extinction of numerous long-lived chondrichthyan (e.g., †hybodontiforms, several groups of lamniforms) and actinopterygian (e.g., †pachycormiforms; †ichthyodectiforms; epipelagic aulopiforms) lineages [1–9]. Most overviews and macroevolutionary studies based on the body and dental fossil record of marine 'fishes' point towards a selective extinction of large-sized and fast swimming apex predators [1–6]. Yet, fossil otoliths —when preserved, or are taxonomically attributable—help detect additional first and last occurrences of major lineages and paint a more complex picture of faunal turnover and extinction patterns of marine teleosts during the K–Pg, by highlighting additional extinction and survival-related variables, pertinent to e.g., life histories, reproductive strategies, and habitat and temperature preferences (summarized in [10]).

This disjunction between studies based on different kinds of datasets is to an extend caused by the poor quality and knowledge of the Maastrichtian–Paleocene body fossil record of marine actinopterygians [1, 3, 11]. Despite few exceptions (e.g., [11, 12]), the fossil 'fish' record from the Maastrichtian Stage (~72–66 Ma) is largely composed of assemblages of disarticulated bones and teeth, of variable systematic informativeness and typically skewed towards heavily ossified elements of mid- and large-sized taxa (e.g., [13–20]). The study of more complete and systematically informative fish fossils from the Maastrichtian is crucial for attaining a better picture of the K–Pg extinction baseline in the marine realm. Renewed collection and research efforts on fossil 'fishes' from the Maastrichtian pelagic marine strata of Pindos Unit in Greece (Fig 1A and 1B) are opening new windows into the pelagic fish biodiversity and ecosystems of the Tethys during the last few million years of the Mesozoic [11, 21].

Gavdos is a small island (~32 km$^2$) that lies near the southern-most tip of the Hellenic Arc, in the Libyan Sea, approximately 38 km to the South of Chania province in Western Crete. Fossil 'fishes' from late Miocene deposits of the island have been known for over a century [23–26]. Recently, another fish assemblage was discovered in mid–late Maastrichtian strata of the Pindos Unit exposed on the western and southern part of the island ([21]; Fig 1C). The latter fossiliferous horizons correspond to marly limestones that have been quarried to produce slabs used as pavestones in at least two locations; the lighthouse (Faros) and Aghia Triada Chapel. The preliminary study of this assemblage was based on a couple of fossil specimens retrieved from pavestones and revealed the presence of two taxa: an †ichthyodectoid of possible †saurodontid affinities and †*Enchodus* cf. *dirus* [21]. The nowadays abandoned quarry (Fig 1D) that produced the pavestones is located near the settlement of Vatsiana (34˚48'56.69"N; 24˚6'21.50"E), but the fossiliferous horizons (Fig 1E) are largely covered in debris.

In the summer of 2019, the authors performed prospecting and small excavations at the Vatsiana Quarry and a careful survey of pavestones in the abovementioned locations, in order to retrieve and rescue additional fossil samples. Slab extraction directly from the fossiliferous horizons did not yield any identifiable remains, possibly due to low fossil concentration, but several identifiable specimens were recovered from the debris of past quarrying activity. Most of the better-preserved specimens were embedded in slabs used as pavestones around the lighthouse. We placed particular emphasis on rescuing all such reasonably-preserved and identifiable remains, which have otherwise been subject to exposure and weathering for over a decade. We hereby provide a detailed description of a fossil sample over 20 identifiable specimens of ray-finned fishes, and one belonging to a shark. The new material allows us to considerably expand the list of taxa known from the Maastrichtian of the Pindos Unit exposures both in Gavdos and as a whole. It also provides a basis for the recognition of two new species of

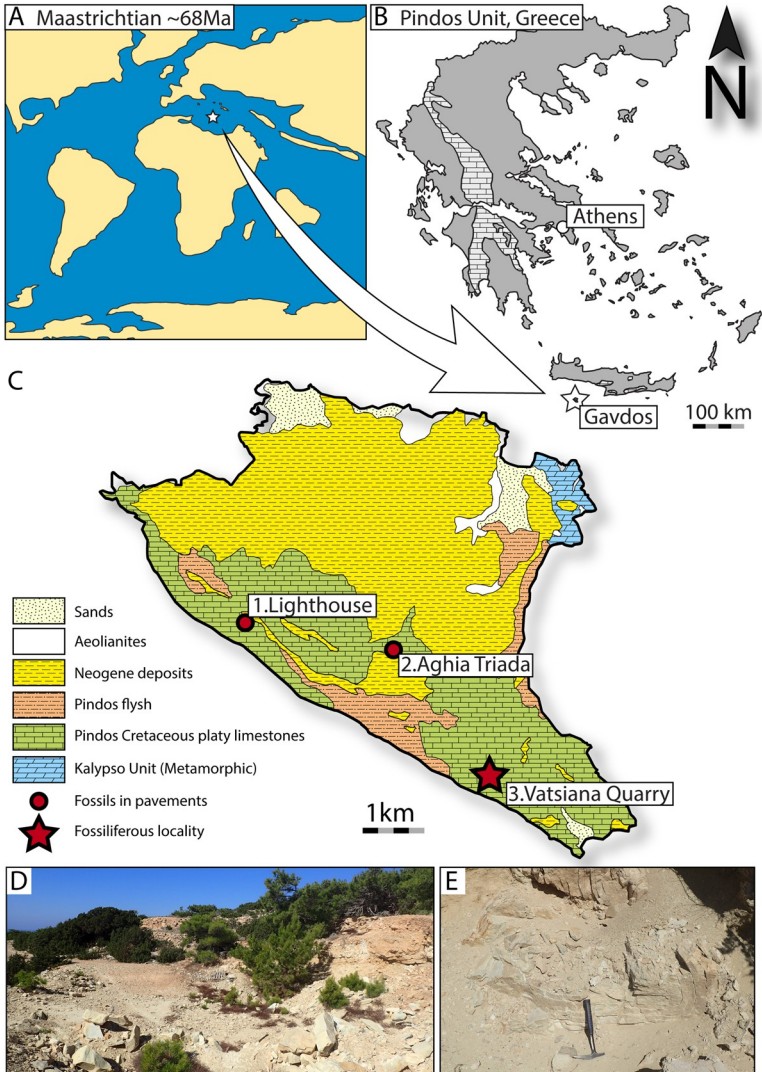

**Fig 1. Geological background and locality information.** (A) Tentative location of the Pindos Unit within the Tethys during the Maastrichtian (Maastrichtian map based on [22]); (B) Map of the main exposures of the Pindos Unit (light grey) in continental and insular Greece; (C) Simplified geological map of Gavdos, showing the main lithologies exposed on the island, and the localities where fossils were found and collected (modified from [21]); (D) overview of the Vatsiana Quarry; (E) fossiliferous marly-limestone horizon exposed in the quarry.

aulopiform teleosts, one of which is accommodated in a new genus. In addition, it helps ameliorate the global, and the Tethyan in particular, deficiency in deep/offshore 'fish' body-fossils from the time interval immediately preceding the K–Pg Extinction.

## Geological background and age of deposits

Despite its small size, Gavdos is characterized by a complex geological setting. Three major series of geological formations are observed (Fig 1):

**Formations of the calypso unit.** The Calypso Unit is exposed on the Northeastern tip of the island and comprises an ophiolithic volcano-sedimentrary metamorphic complex [27] and a transgressional sedimentary nappe of Late Cretaceous age [28]. The metamorphic event recorded in the Calypso Unit took place under high pressure (>7kb) and high temperature

conditions (≈500˚C) during the late Jurassic (~148 Ma; [27]). The Calypso Unit is emplaced as a nappe on top of the Pindos Unit.

**Sedimentary sequences of the pindos unit.** The intensely folded and faulted alpine sedimentary basement of Gavdos dominates the landscape of the western and southern part of the island. It comprises Late Jurassic–middle Eocene sedimentary rocks, which form part of the Pindos Unit [21, 28, 29]. The Pindos Unit corresponds to a series of pelagic sedimentary rocks, which were deposited on the passive margin between the Gavrovo-Tripolitza Platform—a promontory of the Adria microplate—and the Pindos Ocean between the Late Triassic and the Maastrichtian–Paleocene [30, 31]. The rocks of the Pindos Unit are exposed along a North–South Axis starting in Montenegro, and passing through Albania, Central–Western continental Greece and Central Peloponnese, to reach Crete and Gavdos to the South (Fig 1B and 1C). On Gavdos, the oldest lithologies associated with the Pindos Unit are red silts and radiolarites of Late Jurassic age. These are followed by clastic deposits of Cenomanian age known as the 'first flysch of Pindos', and then by thinly-bedded carbonates with occasional silex content, of Late Cretaceous age [21, 28, 29]. The passage from the pelagic carbonate sedimentation to the clastic sedimentation associated with the flysch of Pindos Unit is marked by thinly-bedded marly limestone facies, which dates to the Maastrichtian–Paleocene, with the top-most layers possibly extending into the early Eocene [21, 28, 29]. As in other parts of Greece where the Pindos Unit is exposed [11, 26], the stratigraphically deeper layers of these transitional marly-limestone horizons yield 'fish' fossils [21], including the specimens studied herein. The overlying flysch comprises alternations of arenaceous and pelitic facies, deposited during the early to middle Eocene [21, 28, 29].

**Neogene and younger deposits.** The largest portion of Gavdos is dominated by exposures of Miocene sedimentary sequences, which unconformably overly the alpine basement. These are subdivided into two formations, the shallow-water late Serravalian Potamos Formation (Fm.). and the deep-water Tortonian–Messinian Metochia Fm. [32–34]. The Neogene deposits of Gavdos were subjected to intensive geological, paleoclimatic, and paleontological studies during the past six decades (e.g., [32–41]). Amongst other fossils, the Metochia Fm. hosts moderately diverse teleost assemblages of Tortonian [23] and Messinian [24] age.

The various exposed geological units of Gavdos are at places unconformably overlain by sandy and marly deposits containing Pleistocene–Holocene gastropods [42, 43].

## Age of fossiliferous horizons

The age of the fossiliferous horizons that yielded the 'fish' material examined in this work correspond to the stratigraphically deeper layers of the 'transitional' marly limestone. These were biostratigraphically constrained to the mid–late Maastrichtian, on the basis of contained planktonic foraminifera [21]. More specifically 'fish'-bearing slabs previously collected from the same pavements, and deriving from the same horizons exposed in Vatsiana Quarry, produced foraminiferal associations characteristic of the †*Contusotruncana contusa*–†*Racemiguembelina fruticosa* biozones [21]. This, in conjunction with the absence of biomarkers for the latest Maastrichtian †*Abathomphalus mayaroensis* biozone, roughly indicate an absolute age constraint between 69.18 Ma and 70.14 Ma [21, 44].

## Materials and methods

All fossils were carefully washed with clean water and lightly brushed to remove growing plant matter and dust. Selected fossils with details hidden in the matrix were subjected to topical 5–9% HCL treatments and/or mechanical preparation using a needle and an air-scribe. Following their acid treatment, the fossils were washed with clean water and were subsequently

given a thin coat of acetone soluble Paraloid glue for protection. The fossils described here are cataloged and stored in the collections of the Paleontology and Geology Museum (AMPG) of the National and Kapodistrian University of Athens (NKUA). The letters 'VTS' in the specimen numbers indicate that they derive from the Vatsiana Quarry. In the future, some pavestone slabs with fossils will be loaned to the Municipality of Gavdos for exhibition, upon completion of the local natural and geological heritage information center. Necessary paleontological fieldwork permits were provided by the AMPG and the NKUA. Throughout the text and figures, extinct taxa are preceded by the dagger symbol (†). Actinopterygian classification follows [45].

### Nomenclatural acts

The electronic edition of this article conforms to the requirements of the amended International Code of Zoological Nomenclature, and hence the new names contained herein are available under that Code from the electronic edition of this article. This published work and the nomenclatural acts it contains have been registered in ZooBank, the online registration system for the ICZN. The ZooBank LSIDs (Life Science Identifiers) can be resolved and the associated information viewed through any standard web browser by appending the LSID to the prefix "http://zoobank.org/". The LSID for this publication is: urn:lsid:zoobank.org: pub:406875DB-F286-48B2-8F09-A200AC2E56E7. The electronic edition of this work was published in a journal with an ISSN, and has been archived and is available from the following digital repositories: LOCKSS.

### Institutional abbreviations

**AMPG**: Paleontology and Geology Museum (of the NKUA); **BSPG**: Bayerische Staatssammlung für Paläontologie und Geologie; **MHNG**: Muséum d'Histoire Naturelle Genève; **NKUA**: National and Kapodistrian University of Athens.

### Comparative material studied

*Alepisaurus brevirostris* Gibbs, 1960 [46] (MNHN-IC-2002-1105); *Alepisaurus ferox* Lowe, 1833 [47] (MNHN-IC-2005-0218); †*Nematonotus longispinus* (Davis, 1887) [48] (MHNG-GEPI-V891b; MHNG-GEPI-V892b); †*Sardinioides monasteri* (Agassiz, 1835) [49] (BSPG_AS_-VII877; BSPG_AS_VII878)

## Results

### Systematic paleontology

Actinopterygii *sensu* Goodrich, 1930 [50] and [51]

Teleostei Müller, 1845 [52]

†Ichthyodectiformes Bardack and Sprinkle, 1969 [53]

†Ichthyodectoidei Romer, 1966 [54]

†Ichthyodectoidei indet.

Fig 2A–2D

**Material.** AMPG_VTS_7, scale; AMPG_VTS_8, scale; AMPG_VTS_13, scale; AMPG_VTS_21, scale; AMPG_VTS_22, scale; AMPG_VTS_29, caudal fin

**Description.** *Caudal skeleton and fin*. The posteriormost portion of the axial skeleton and caudal fin are preserved in this fossil. The vertebrae are approximately as long as tall and bear autogenous neural and haemal arches and spines. Five to six vertebral elements are involved in the support of the caudal fin rays. We identify at least one ventral hypural (H1). The latter

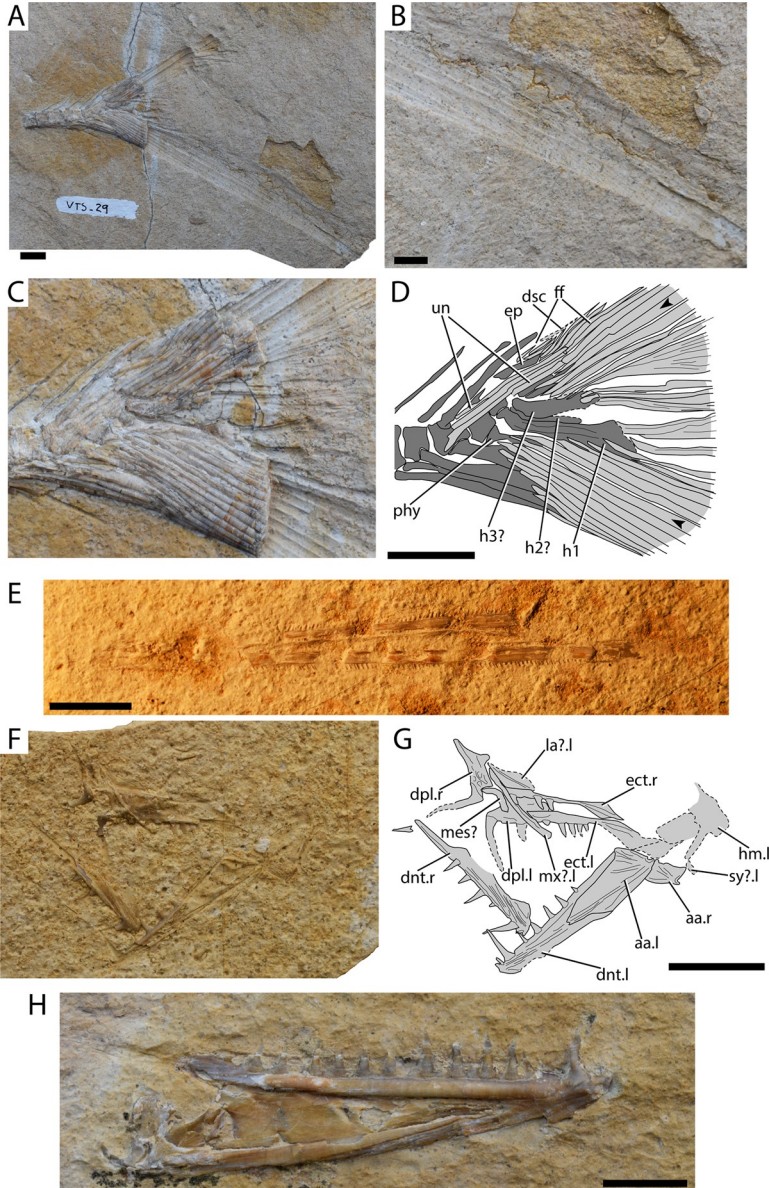

**Fig 2. †Ichthyodectoidei, †Dercetidae and †Enchodontidae from Gavdos.** (A) †Ichthyodectoidei indet.
AMPG_VTS_29, caudal skeleton and fin; (B) Magnification of ventral caudal fin lobe of A; (C) Detail of the caudal
skeleton of A; (D) interpretative drawing of C; (E) †Dercetidae indet. AMPG_VTS_27, lower jaws; (F) †Enchodontidae
indet. morphotype 1 AMPG_VTS_20, palate, lower jaws and associated elements; (G) interpretative drawing of F; (H)
†Enchodontidae indet. morphotype 2, lower jaw in medial view. Abbreviations: aa. anguloarticular; dnt. dentary; dpl.
dermopalatine; dsc. dorsal scute; ect. ectopterygoid; ep. epural; ff. fringing fulcra; h1–3. hypural 1–3; hm.
hyomandibula; la? putative lachrymal; mes? putative mesethmoid; mx. putative maxilla; phy. parhypural; sy.
symplectic; un. uroneurals;.l. left-side indicator;.r. right side indicator. Black arrowheads indicate outermost principal
caudal fin rays. Scale bars equal 1 cm.

bears a thickened anteroventrally to laterally running ridge. A possible second small and slender ventral hypural (H2), lying immediately dorsal to H1, is tentatively deduced based on its imprint on the matrix. Two to three dorsal hypurals are visible, bearing branched rays associated with the dorsal lobe. Additional hypurals are likely present but obstructed by the uroneurals and fin rays. The articular head of the first dorsal hypural (H3) is slightly curved dorsally.

Three or four uroneurals are preserved, but their count cannot be securely established. Their anterior tips overlap the lateral surfaces of the first two preural centra. A putative tiny scute is seen in the anterior base of the dorsal lobe. Up to nine procurrent rays are associated with the dorsal lobe, but some of them cannot be easily differentiated from conceived epurals. The relevant count for the ventral lobe cannot be approximated, due to lack of preservation. The principal rays of both lobes bear numerous, closely packed oblique segmentations. Each lobe comprises seven to eight branched (excluding the principal) rays. The rays of the middle portion of the caudal fin, as well as the ventral rays of the dorsal lobe are shorter and show stronger branching. Two such branched rays are associated with H1, while two to three are associated with the putative H3. The distal portions of the dorsal and ventral lobes bear characteristic oblique segments, as evidenced by their imprints. A striking feature of this fossil relates to the ventral lobe of the tail fin, which is at least one and a half times longer and more robust than the dorsal lobe.

*Scales.* The scales are rounded sub-quadrangular to sub-quadrangular in shape. They are taller than long. The focal point is elliptically-shaped and located in the center of each scale. The scale area between the focal point and outer scale margins is covered in densely-packed circuli. The anterior field bears numerous (>11) radii, which radiate from the focal point.

**Remarks.** The fossil in question can be attributed to †Ichthyodectoidei on the basis of similarities with the group, rather than recognized synapomorphies [55]. These similarities include: i) presence of autogenous neural and haemal spines, which are strongly bent posteriorly; ii) angled proximal extremity of second hypural; iii) strengthened caudal fin base formed by tightly packed uroneurals and fin rays; iv) strong asymmetry between the two caudal lobes, with the ventral one being much longer than the dorsal; v) oblique segmentation of the caudal rays with a step-like pattern; vi) relatively short uroneurals, not exceeding preural centrum 2; vii) scales bearing anterior radii (see also [55, 56]). The fifth character sets AMPG_VTS_29 apart from known Campanian–Maastrichtian †ichthyodectiform genera (for stratigraphic ranges see [55]), such as †*Saurodon* [57]; †*Xiphactinus* [58] and †*Gillicus* [55]. The caudal fins of †*Saurocephalus* and †*Ichthyodectes* are still poorly known and it cannot be excluded that this material is referable to one of these genera [55, 57, 59]. Cavin et al. [21] described a caudal fragment of an indeterminate †ichthyodectoid from the same horizons in Gavdos, which exhibits a dorsal and anal fin pattern similar to †*Saurodon elongatus* from the Late Cretaceous of Nardò [57]. As noted above, †*S. elongatus* has symmetrical caudal lobes [57] distinguishing it from AMPG_VTS_29. The well-developed ventral caudal fin lobe of AMPG_VTS_29 is instead reminiscent of that of the Cenomanian †*Eubiodectes* [55]. It is therefore likely that at least two different species of †ichthyodectoids are present in the Gavdos assemblage. The isolated scales do not provide additional taxonomic information, but resemble in most regards †ichthyodectoid scales from other Campanian–Maastrichtian localities [12, 59, 60].

Eurypterygii Rosen, 1973 [61]

†Dercetidae Pictet, 1850 [62]

†Dercetidae indet.

Fig 2E

**Material.** AMPG_VTS_27, lower jaw with teeth

**Description.** Almost the complete length of both dentaries is preserved, but the proximal portion of the jaw is missing. The dentaries are straight and slender, tapering only slightly towards the symphysis. The anterior tip of the symphysis is rounded. The dentaries bear tiny, equally spaced, tiny conical teeth along their length. The teeth are slightly inclined posteriorly.

**Remarks.** This material can be readily distinguished from the longirostrine †'ichthyotringoid' studied here, since the latter exhibits a marked degree of heterodonty. Despite being incomplete, this jaw lacks a pre-symphyseal dentigerous portion, and can thus be securely

excluded from e.g., the longirostrine †aspidorhynchiforms and is, thus, better assigned to a †dercetid. The length and slenderness of the jaw are reminiscent of markedly longirostrine Late Cretaceous †dercetids, such as †*Apuliadercetis* [63]. Similar jaws assigned to †Dercetidae were recognized in the late Maastrichtian of Pindos Unit in Eurytania, continental Greece [11, 64].

†Enchodontidae Woodward, 1901 [65]

†Enchodontidae indet. morphotype 1

Fig 2F and 2G

**Material.** AMPG_VTS_3a,b, damaged dermopalatine with fang; AMPG_VTS_10, quadrate and ectopterygoid; AMPG_VTS_20, lower jaw and palatine fragments

**Description.** Aspects of both palatines and lower jaws, as well as the posterior portion of the suspensorium of a small individual are preserved in AMPG_VTS_20, rendering it the most complete fossil of this morphotype.

*Palate and associated ossifications.* The dermopalatine is longer than tall and bears a thin, unornamented and slightly posteriorly curved fang. A tall plate on the dorsal margin of the dermopalatine is situated immediately posterior to the level of the fang. The ectopterygoid is a slender and elongate bone. The posterior portion of the ectopterygoid widens dorsoventrally and forms an acute tip. Approximately six, laterally compressed and slightly posteriorly inclined teeth are born by the ectopterygoid. The ectopterygoid teeth become gradually smaller posteriorly. The quadrate forms a convex anterior margin, and extends ventral to the level of the metapterygoid. Traces of the metapterygoid, the hyomandibula and the symplectic are observed, but they are poorly preserved and their outlines could not be accurately traced.

*Jaws.* An edentulous, weakly curved element overlaps the left dermopalatine. We tentatively identify this element as the maxilla. An additional smaller and wider element is situated dorsally to anterior tip of the putative maxilla and bears a trace of a crest or canal extending along most of its length. This is tentatively identified here as a lachrymal. The left lower jaw is almost completely preserved. The anguloarticular is almost rhomboidal, forming a well-developed, triangular anterior process, which articulates with the dentary. The articular fossa is situated on the posteroventral tip of the anguloarticular and opens posterodorsally. Traces of striae are observed on the lateral surface of the bone, radiating from the articular fossa. The dentary bears a deep, V-like concavity for the insertion of the anguloarticular. Its dorsal margin is straight, while the ventral one is incompletely preserved and could not be traced. The dentary forms a ventral expansion near the symphysis. The lateral surface of the dentary is ornamented with densely packed striae. A tall, straight fang is observed immediately behind the symphysis, and is preceded by at least one tiny tooth. Five smaller, well-spaced and weakly posteriorly inclined fangs are observed behind the main fang, but additional fangs might have been present but not preserved.

AMPG_VTS_3a,b and AMPG_VTS_10 are poorly preserved but are tentatively included in the same morphotype on the basis of similarities in dermopalatine fang morphology and in the number of ectopterygoid teeth, respectively, with AMPG_VTS_20.

**Remarks.** We recognize clear similarities—not synapomorphies—between this material and members of †Enchodontidae: i) presence of a single fang on the dermopalatine; ii) ectopterygoid bearing between six to eight sparsely arranged teeth; iii) lower jaw with low coronoid process; iv) presence of a long near-symphyseal fang on mandible; v) striated ornamentation of the lateral surface of the mandible (see e.g., [12, 13, 18, 19, 21, 66–69]). The morphology of the dermopalatine and its teeth, and to a lesser degree, that of the ectopterygoid and lower jaw bones and fangs are historically treated as a basis for distinguishing among different species of †enchodontids, even when fragmentary remains are considered (e.g., [13, 18, 19, 21, 66, 67]). This approach, especially in the case of latest Cretaceous occurrences, has led to the possible

lumping of †enchodontid species under the commonly recognized genus †*Enchodus* [21]. The phylogenetic diagnoses of the group comprising †*Enchodus*-like fishes were volatile in recent analyses, and relied significantly on homoplasic characters, many of which pertain to anatomical details not preserved in our material (e.g., sensory canals; postcranium; squamation; see [68–72]). This fact, in conjunction with the poor state of preservation of fine dental details in the Gavdian fossils (unclear extent of cutting edges or outline of cross-section of the dermopalatine fang), prevent us for securely attributing it to a genus or species. However, we note that this morphotype is best excluded from the previously recognized †*Enchodus* cf. *dirus* [21], as it seems to lack tooth barbs or strongly recurved dentary teeth. †Recently described †enchodontids from the late Maastrichtian of Pindos Unit in Eurytania, Greece, exhibit broader teeth on the palate and lower jaw but also derive from larger individuals [11]. The possible ontogenetic transformation of the dentition is unknown in †enchodontids.

†Enchodontidae indet. morphotype 2

Fig 2H

**Material.** AMPG_VTS_28, lower jaw

**Description.** The medial surface of the lower jaw AMPG_VTS_28 is visible on the specimen. The jaw is rather shallow, forming a straight occlusal margin and a gently convex ventral margin. The anguloarticular seems to be restricted on the posterior sixth of the mandible, with its articular fossa situated on the posteroventral tip of the bone and opening dorsally. The dentary is elongate, tapering anteriorly and seemingly forming a very shallow symphysis. Anteriorly, a recurved fang is present, stemming from a thick base, and is preceded by a smaller symphyseal tooth. A small gap separates the main fang from smaller, regularly spaced fangs. At least 12 teeth posterior to the main fang are preserved. Anteriorly, these fangs are thinner and recurved, but become shorter and stockier posteriorly. All teeth are missing their apices and as a result the possible presence of barbs cannot be established.

**Remarks.** This mandible bears more teeth than the previously described †Enchodontidae indet. morphotype 1. These teeth are additionally recurved unlike in AMPG_VTS_20. AMPG_VTS_28 differs from the mandible of †*Enchodus* cf. *dirus* previously described from the same deposits [21] in the following features: i) being shallower; ii) forming a straight oral margin; iii) presenting a higher number of more closely spaced teeth; iv) teeth lacking barbs. Recently described †enchodontids from the late Maastrichtian of Eurytania, continental Greece, also lack barbs and present closely spaced fangs [11]. However, the only known lower jaw from Eurytania exhibits more massive, blade-like teeth. It is thus possible and probable that multiple species of †*Enchodus*-like fishes shared habitats in the late Maastrichtian of Greece. AMPG_VTS_28 presents similarities in its overall short and elongate mandibular geometry, the markedly short symphysis and the number of teeth (13) born with †*Enchodus tineidae* from the Campanian of Egypt [71]. We prefer to not assign this material to the latter species, since we cannot observe the ornamentation and the condition of the mandibular canal on the lateral surface of the bone, which is embedded in the matrix.

†*Calypsoichthys* gen. nov.

urn:lsid:zoobank.org:act:E5ABB350-99D7-47AB-AFB8-1F21B60C24E5

*Type species.* †*Calypsoichthys pavlakiensis*

*Diagnosis.* An †enchodontid teleost, exhibiting the following character combination: poorly differentiated 'coronoid' process; a single row of densely packed to overlapping, falcate to triangular teeth on the ectopterygoid and the middle and posterior portion of the dentary; striated ornamentation of lower jaw bones; traces of large and elongate tubercles on opercle, which radiate its center of ossification.

*Etymology.* The name of the genus is derived from the Homeric nymph Calypso (Καλυψώ) and the Greek word for fish, ichthys (χθς). In Homer's Odyssey, Calypso charmed Odysseus

and detained him in her Island, Ogygia, for seven years. Although Gavdos is only contestably identified as the Homeric Ogygia, one easily becomes captivated by the serene atmosphere of this remote island.

†*Calypsoichthys pavlakiensis* sp. nov.

urn:lsid:zoobank.org:act:E74FD159-30BE-4C7E-B623-FACFC306F7E5

*Holotype*. AMPG_VTS_25, slab preserving both palates, the left lower jaw and traces of the opercle of the same individual.

*Etymology*. In honor of the Greek vertebrate paleontologist and now retired professor of the National and Kapodistrian University of Athens, Dr. Parisis Pavlakis, for his contributions in the dissemination of paleobiology.

*Type locality and age*. Marly limestones of the Pindos Unit exposed in Vatsiana Quarry (34˚ 48'56.69"N; 24˚6'21.50"E), Gavdos Island, southern Greece. The age of the fossiliferous horizons was biostratigraphically constrained to the mid–late Maastrichtian, on the basis of planktonic foraminifera [21].

*Diagnosis*. Single species; as for genus.

Fig 3

**Material.** See holotype information above.

**Description.** *Palate and associated ossifications*. Both the left and right palates are partially preserved. The dermal portion of the palate is formed by two bones, the dermopalatine anteriorly and the ectopterygoid posteriorly. Both elements bear teeth. The dermopalatine is the smallest of the two bones. It is dorsoventrally wider than the ectopterygoid, as it forms a posterodorsal process, which possibly served for articulation with the braincase. Posteriorly, the palatine tapers to form a process, which inserts in a corresponding furrow of the ectopterygoid. A single unornamented fang is born by the palatine. The cutting edges of the fang are weakly sigmoidal and lead to an acute tip.

The ectopterygoid is markedly short and rostrocaudally elongate, with its overall shape appearing sigmoidal. Its dorsal margin forms a concavity at mid-length, with the corresponding portion of its ventral surface being convex. A thin flange on the anterior portion of the ectopterygoid produces dorsally to meet the palatine. The posterior tip of the ectopterygoid thickens to form an edentulous almond-shaped process. The ventral surface of the ectopterygoid bears a single row of approximately 22 laterally compressed and tightly packed teeth, giving the bone a saw-like appearance (and conceived function). The ectopterygoid teeth are falcate, with the anteriormost being almost as long as the palatine fang, but the dentition gradually loses height posteriorly. The cancellous base of the teeth forms as an anterior indentation, or constriction. There is some overlap between ectopterygoid teeth, with the posterior cutting edge of each tooth being overlapped labially by the anterior edge of its immediately posterior tooth. A thin ovoid bone—partially preserved as an imprint on the matrix—occupies the dorsal concave dorsal and medial part of the ectopterygoid, and likely represents the endopterygoid. Another shard-like portion of a bone is present dorsal to the right ectopterygoid, but is possibly dislocated. We tentatively identify it as a dislocated metapterygoid, or alternatively the posterior portion of the ectopterygoid.

*Lower jaw*. The lower jaw forms a gentle dorsal curve and is about five times longer than high. The largest bone of the lower jaw is the dentigerous dentary, which is preserved in its entirety. The anterior tip of the dentary is blunt and does not form symphyseal processes. The sensory canal likely rested in a furrow that extends along the ventrolateral surface of the dentary, and posteriorly into the retroarticular process. The coronoid process is incompletely-preserved but it appears to have been poorly developed. The anterior process of the anguloarticular extends into the posterior third of the dentary. The lateral surface of both the dentary and the anguloarticular is ornamented with anteroposteriorly elongate ridges and pits.

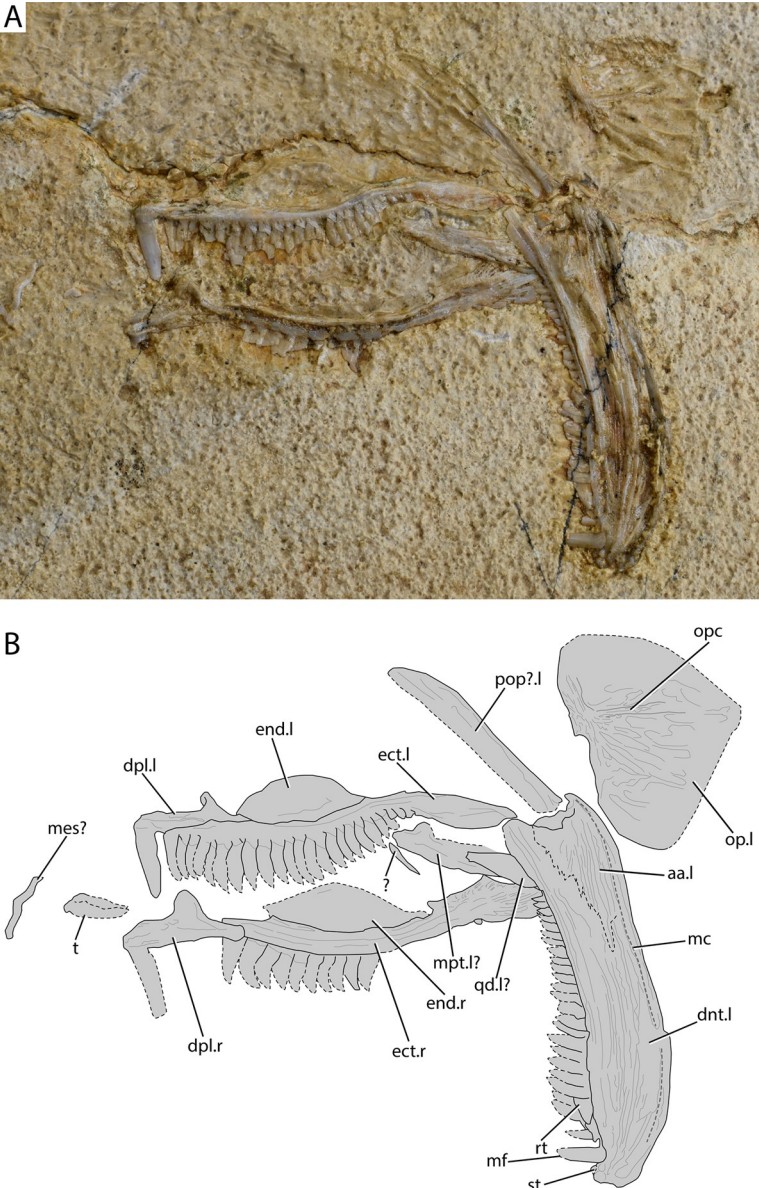

**Fig 3. †*Calypsoichthys pavlakiensis* gen. et sp. nov. from Gavdos.** (A) Lateral view of palates, left lower jaw and associated elements of holotype AMPG_VTS_25; (B) interpretative drawing of A. Abbreviations: aa. anguloarticular; dnt. dentary; dpl. dermopalatine; ect. ectopterygoid; end. endopterygoid; mc. Inferred trace of mandibular sensory canal; mes? putative mesethmoidal fragment; mf. main fang of the dentary; mpt? putative metapterygoid; op. opercle; opc. opercular crest; pop?. putative preopercle; qd? putative quadrate; rt. possible replacement tooth; st. near-symphyseal tooth stub; t. dislocated teeth;.l. left side indicator;.r. right side indicator. Scale bar equals 1 cm.

The articulation with the quadrate is concealed by dermal bone. The dentary bears a single row of teeth. Stubs of up to two tiny teeth occupy the near-symphyseal region and are succeeded immediately posteriorly by a large, weakly sigmoidal fang. A row of ~25 teeth, similar to those of the ectopterygoid, extends from the fang, along the dorsal surface of the dentary, to reach the posterior fifth of the bone. Although anterior 'post-fang' teeth are clearly falcate, posterior teeth are almost triangular. A possible replacement tooth, with its apex pointing anteriorly, is observed almost immediately behind the main fang.

*Additional ossifications*. A damaged bony strut extends obliquely from almost the articulation surface of the anguloarticular. This probably corresponds to the preopercle, or less likely the ventral arm of the hyomandibula, or the posterior margin of the quadrate. We note that we could not observe any sensory openings on the latter bone. The opercle is partially preserved, but its outline could be tentatively reconstructed on the basis of its imprint. Its anterior margin is rather straight. Its dorsal–posterodorsal margin is convex and ends at an acute edge. The posteroventral–ventral margin cannot be securely reconstructed, as the imprint might also include the subopercle. If so, the separation between the two is unclear. A horizontal bony strut/crest crosses the opercle from its point of articulation to its posterior acute edge. The opercle is ornamented with elongate pits and ridges radiating from the articular facet.

**Remarks.** The fossil described herein presents two striking anatomical similarities with †enchodontoid aulopiforms: i) presence of a single palatine fang; ii) presence of a horizontal crest of the opercle [21, 66–68]. A single dermopalatine fang was also described in †*Rharbichthys* [73, 74] and the †dercetid †*Ophidercetis* [75], while a horizontal crest on the opercle is a common feature of other aulopiforms, such as †'ichthyotringoids' [66, 68]. However, a combination of these two characters is only found in members of †Enchodontidae [66–70, 76] and points towards the inclusion of this peculiar fish from Gavdos within the family. The closely spaced, saw-like dentition of the fossil in question differs from that of other †enchodontids from Gavdos [21], or any other known †enchodontid [66, 67, 69, 70, 76–78]. It should, thus, be accommodated in a new genus and species.

Due to its anatomical incompleteness, †*Calypsoichthys* cannot be extensively scored and analyzed phylogenetically, but its preserved aspects can support a preliminary discussion of its affinities through a comparison with known †enchodontid genera. The presence of a single large fang on the palatine characterizes most †Enchodontidae [21, 66–68], to the exclusion of the basally diverging (sensu [70]) †*Unicachichthys* (which bears multiple small teeth and small fangs, [77]), †*Veridagon* (which bears two fangs, [69]), and possibly the †eurypholin †*Vegrandichthys* (it likely bears more than one dermopalatine tooth [70]). Amongst more derived †enchodontids, we note similarities in the ridged ornamentation of the lower jaw bones of †*Calypsoichthys* with †enchodontins—such as †*Enchodus* [21, 66, 67]. The opercle on the other hand bears traces of elongate tubercles and an acute posterior margin, which better resemble those of †eurypholin †enchodontids [66, 67]. The densely packed teeth on the lower jaw and ectopterygoid of †*Calypsoichthys* resemble the dentition on the posterior portion of the dermopalatine—but not the reduced ectopterygoid—and lower jaw of modern alepisaurid aulopiforms (pers. obs. by TA on *Alepisaurus ferox* and *A. brevirostris*). We interpret this likely convergence as a possible adaptation for shearing soft prey. Bone fragments exhibiting identical dentitions to those of †*Calypsoichthys* were recovered from penecontemporaneous exposures of the Pindos Unit in Eurytania, continental Greece ([11]: Fig 6E and 6F), leading us to also attribute them to this newly erected genus.

†Ichthyotringidae Jordan, 1905 [79]

†*Ichthyotringa* Cope, 1878 [80]

†*Ichthyotringa pindica* sp. nov.

urn:lsid:zoobank.org:act:A678E137-9924-4B32-8910-06F39AF6D131

*Holotype*. AMPG_VTS_30

*Paratypes*. AMPG_VTS_2, anterior fragments of rostrum and lower jaw; AMPG_VTS_15, fragmentary mandibles with teeth; AMPG_VTS_17, fragmentary rostrum; AMPG_VTS_26, rostrum and lower jaw.

*Etymology*. The specific name †*I. pindica* reflects the distribution of this species in the Maastrichtian of the Pindos Unit in Greece.

*Type locality and age*. Same as for †*Calypsoichthys pavlakiensis* gen. et sp. nov. above.

*Diagnosis.* Distinguished from all other species of the genus by the presence of a weakly-developed coronoid process and its unique dentition on the dermopalatine and the dentary, which includes numerous anteriorly inclined main fangs, flanked by small conical teeth.

Fig 4

**Material.** See information on type series above.

**Description.** The skull AMPG_VTS_30 (Fig 4A and 4B) comprises most of the anatomical regions preserved in other specimens and provides a good base for describing this taxon. It lacks the opercular series and possibly some ossifications of the upper jaw (maxillae,

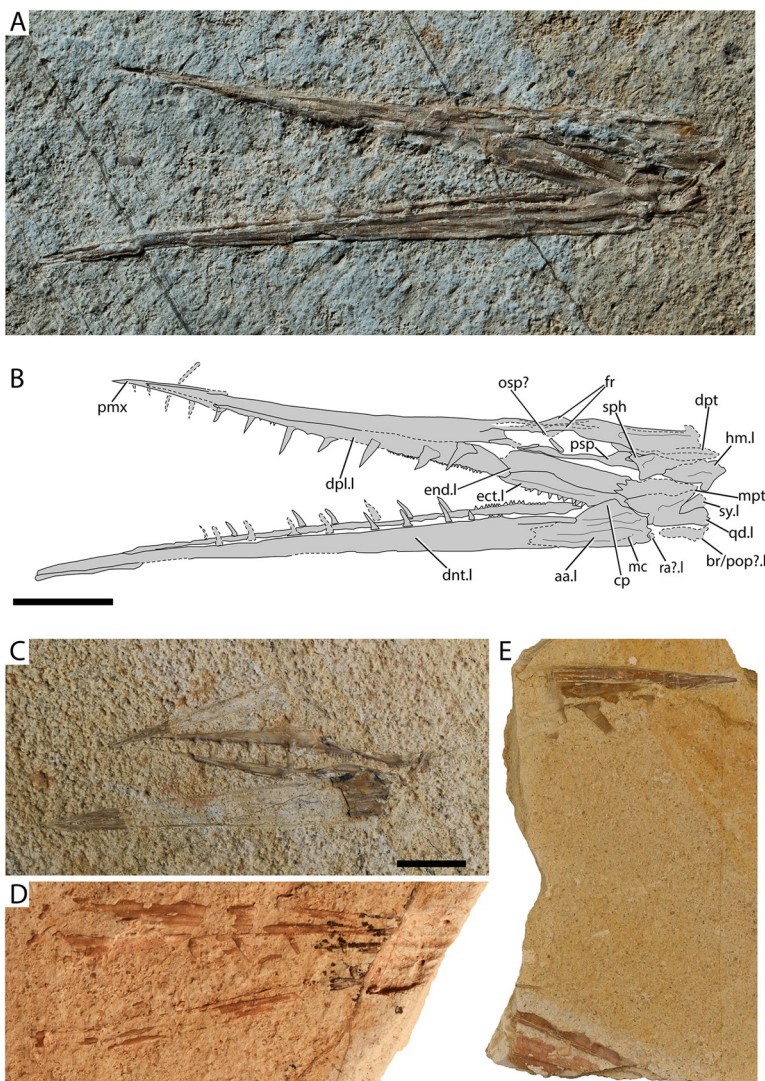

**Fig 4. †*Ichthyotringa pindica* sp. nov. from Gavdos.** (A) Lateral aspect of the skull and mandible of the holotype AMPG_VTS_30; (B) interpretative drawing of A; (C) dermal palate, mandible and rostral imprint of designated paratype AMPG_VTS_26; (D) rostral and lower jaw fragments with teeth of AMPG_VTS_15; (E) rostral and mandibular fragments of AMPG_VTS_2. Abbreviations: aa. anguloarticular; br/pop? trace of putative branchiostegal ray or preopercle; cp. coronoid process of anguloarticular; dnt. dentary; dpl. dermopalatine; dpt. dermopterotic; ect. ectopterygoid; end. endopterygoid; fr. frontals; hm. hyomandibula; mc. trace of mandibular canal; mpt. metapterygoid; osp? trace of putative orbitosphenoid; pmx. premaxilla (fused); psp. parasphenoid; qd. quadrate; ra? trace of putative retroarticular; sph. sphenotic ossification (autosphenotic); sy. symplectic;.l. left side indicator;.r. right side indicator. Scale bar equals 1 cm.

supramaxillae). The preorbital region of the skull and the mandible form a prominent, acute rostrum. The estimated skull length is 6.13 cm (measured from the tip of the rostrum to the conceived posterior border of the dermopterotic). The preorbital region of the skull (measured from the anterior thickening of the parasphenoid) is 4.43 cm in length, which is ~72% of the skull length estimate. The articulation of the lower jaw is situated immediately posterior to the level of the orbit.

*Cranium.* The skull roof is rather poorly preserved, and the boundaries of individual ossifications could not be traced. The skull roof bones lack any ornamentation. A V-like element—possibly formed by the fused premaxillae—forms the tip of the rostrum, and bears traces of two ventrally–posteroventrally extending fangs. The constituents of the dorsal surface of the rostrum could not be traced, but a median elongate mesethmoid ossification was likely present. The frontals are narrow and markedly elongate, reaching into the at least the posterior third of the rostrum. Traces of the supraorbital sensory canals can be seen extending anterior to the level of the orbit and directly above it. The dorsal orbital margin is marked by a weak indentation of the frontals. Traces of an infilled sensory canal directly above the hyomandibula give away the presence of an elongate but thin dermopterotic. A possible epiotic occupies the posterodorsal corner of the specimen. The parasphenoid is rather straight, forming a posteriorly inclined and short ascending process at the level of the posterior corner of the orbit. Anteriorly, at the conceived anterior margin of the orbit the parasphenoid forms a dorsal convexity, reaching towards the conceived orbitosphenoid. A thick, laterally projecting ossification on the posterodorsal portion of the parasphenoid likely corresponds to the postorbital process of the autosphenotic.

*Palate and suspensorium.* Five ossifications of the palatal apparatus are identified. The most prominent of latter is the anteriorly-situated dermopalatine, which roughly forms the posterior two-thirds of the rostrum. The dermopalatine is much longer than high and its anterior tip is tapered, conforming to the shape of the rostrum. The posterior margin of the dermopalatine is rounded and forms a posteroventral tip, which inserts between the ectopterygoid and the endopterygoid. The dorsal and ventral margins of the dermopalatine are straight. The dermopalatine bears approximately six prominent and anteriorly tilted, and widely spaced fangs. Imprints of tiny teeth are observed between fangs, on the ventrolingual surface of the bone. Posterior to the palatine lies the ectopterygoid. The ectopterygoid is shallow and elongate, exhibiting tapering anterior and posterior margins. It bears small, spaced conical to triangular teeth, which resemble the ones on the posterior portion of the dentary. The endopterygoid is elongate and smooth, and lies directly dorsal to the ectopterygoid. The metapterygoid is the smallest bone of the palatoquadrate series. It is anteroposteriorly elongate and is wedged among the endopterygoid, the ectopterygoid and the 'dorsal' margin of the quadrate. The anteroventral margin of the quadrate is straight, but is deeply indented immediately anterior to the condyle of the quadrate. A robust symplectic lies within a notch on the posterodorsal margin of the quadrate. Ventral–posteroventral to the quadrate there are traces of a narrow and elongate dermal ossification, which could either correspond to a branchiostegal ray, and/or the ventral limb of the preopercle. The hyomandibula is rostrocaudally elongate, forming a wide anterior plate. The ventral limb is incompletely preserved, but must have not been well-developed. The opercular process is small and poorly differentiated.

*Lower jaws.* The mandible reaches slightly further anteriorly than the preorbital region of the skull. The quadrate-anguloarticular articulation occurs near the posteroventral margin of the mandible, with the anguloarticular fossa being directed posteriorly–posterodorsally. A distinct but shallow triangular coronoid process is formed by the anguloarticular. Two thickened ridges radiate from the articular fossa of the anguloarticular. One extends anterodorsally and delimits the posterior margin of the coronoid process. The other ridge extends anteriorly. The

suture between the anguloarticular and the dentary could not be traced. The sensory canal was probably accommodated in a furrow along the ventral portion of the dentary. The dentary is dentigerous along its length, with tiny, closely-spaced teeth being present on the proximal quarter of the bone, giving way to large, rostrally inclined fangs distally. There are no small teeth between the fangs of the dentary. The fangs of the dentary bear a robust and somewhat wide base. Faint apicobasally running striations (plicidentine?) are observed immediately distal to the attachment base of some of the fangs. All tips are weathered, thus, lacking their acrodine caps. Posterior dentary teeth are similar in structure to the fangs, but also exhibit a short acrodine cap (roughly 1/3$^{rd}$–1/4$^{th}$ of tooth lenth). The remaining specimens (Fig 4C–4E) confirm the anatomical observations described above. AMPG_VTS_2 (Fig 4E) is of particular interest, as it comes from a much larger individual than AMPG_VTS_30.

**Remarks.** The material presented here is provisionally attributed to the Late Cretaceous '†ichthyotringoid' (sensu [66]) †*Ichthyotringa* on the basis of a combination of anatomical features shared with species assigned to the genus [49, 66, 68, 74, 78, 81–85]. These similarities are not formally recognized as synapomorphies but include: i) markedly elongate preorbital region, forming an acute rostrum; ii) presence of a small, median dentigerous premaxilla capping the snout; iii) dermopalatine being the principal and largest tooth-bearing bone of the 'functional upper jaw'; iv) presence of a single row of teeth on the lower jaw and palate; v) unornamented dermal bones of the skull roof. Many of the characters that are employed to diagnose the genus †*Ichthyotringa* refer to its postcranial anatomy (e.g., head to body ratio; caudal fin anatomy; see [66, 68]) and are not preserved in the fossil material found in Gavdos.

The fossils in question also somewhat resemble the Campanian–Maastrichtian '†ichthyotringoid' †*Apateodus* [12, 48, 66, 86, 87] in exhibiting a 'coronoid process' of the anguloarticular. However, the Gavdos †ichthyotringids can be easily differentiated from †*Apateodus*, as the former exhibit a longer rostrum; a dermopalatine almost three times longer than the ectopterygoid; a longer and slenderer mandible; the presence of a higher number of mandibular fangs, all of which appear to not be flattened or striated. The genus †*Apateopholis* shares a forwardly inclined suspensorium with the Greek fossils in question, but differs from them in exhibiting a larger premaxilla; a markedly deeper orbital–postorbital region; a taller palate; a dentition comprising only tiny teeth and by the tubercular ornamentation of its skull roof. Finally, the much larger †*Ursichthys*—the final remaining taxon associated with '†ichthyotringoids'—is largely known from partial neurocranial and pectoral-girdle elements [87], which cannot be sufficiently compared with the material in question. Its hyomandibula however does not form an anteriorly inclined ventral limb [87]. We note that the monophyly of Goody's [66] '†ichthyotringoidei' is disputed by recent phylogenetic works [68, 72].

Previous to this work, there was a total of seven nominal species assigned to the genus †*Ichthyotringa* [48, 49, 66, 74, 78, 81, 83–85], none of which was known to reach the Maastrichtian Stage. The anteriorly inclined direction of the fangs on the palate and lower jaw for the Gavdian specimens, as well as their 'coronoid process', differentiate it from all other species of the genus and, thus, warrant the establishment of a new taxon to accommodate it. We note that numerous specimens preserving partially articulated or disarticulated crania and jaws of what appears to be the same species of †*Ichthyotringa* have been reported from the late Maastrichtian of Pindos Unit in Eurytania, continental Greece [11, 64]. The †*Ichthyotringa* material from continental Greece is identical in all aspects to that from Gavdos, and further exhibits a horizontal bar in the opercle [11].

Eurypterygii incertae sedis
†Sardinioididae Goody 1969 [66]
cf. †Sardinioididae indet.
Fig 5A–5C

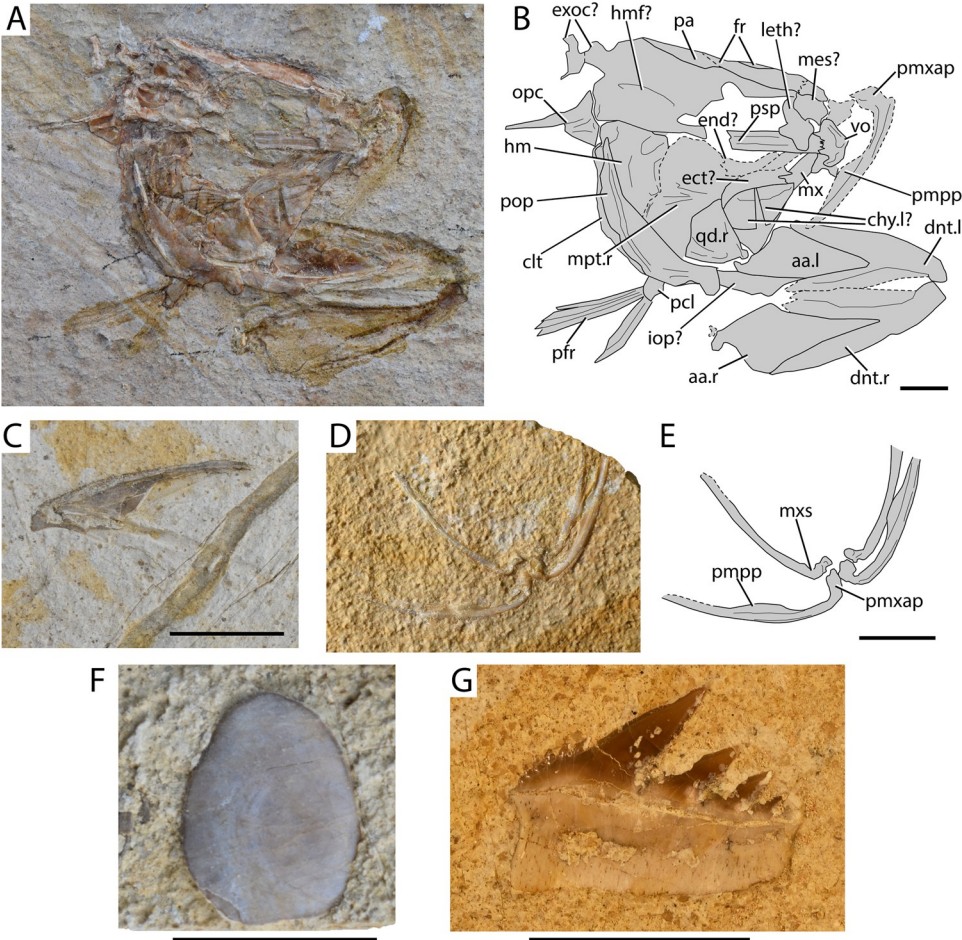

**Fig 5. cf. †Sardinioididae, unidentified teleosts and †*Gladioserratus* sp. from Gavdos.** (A) partial cranium of cf. †Sardinioididae indet. AMPG_VTS_31; (B) interpretative drawing of A, with dotted lines indicating tentative reconstructions of bone boundaries; (C) cf. †Sardinioididae indet. AMPG_VTS_5, isolated dentary tentatively ascribed to the same morphotype as A; (D) Eurypterygii indet. AMPG_VTS_16, upper jaw bones; (E) interpretative drawing of D; (F) Teleostei indet. AMPG_VTS_9, scale possibly deriving from an †ichthyodectiform; (G) †*Gladioserratus* sp. AMPG_VTS_1, lower? anterior tooth. Abbreviations: aa. anguloarticular; chy. ceratohyal; clt. cleithrum; dnt. dentary; ect? putative ectopterygoid; end? putative endopterygoid; exoc? putative dislocated exoccipitals; fr. frontals; hm. hyomandibula; hmf? putative hyomandibular facet; iop? putative interopercle (fragment); leth? putative lateral ethmoid; mes? putative mesethmoid ossification; mpt. metapterygoid; mx. maxilla; opc. remnant of opercular crest; pa. parietal; pcl. postcleithrum; pfr. pectoral fin rays; pmpp. postmaxillary process of premaxilla; pmxap. anterior process of premaxilla; pop. preopercle; psp. parasphenoid; qd. quadrate; vo. vomer;.l. left side indicator;.r. right side indicator. Scale bars equal 1 cm.

**Material.** AMPG_VTS_5, isolated dentary; AMPG_VTS_31, skull and jaw bones

**Description.** The skull (Fig 5A and 5B) is strongly compressed laterally and damaged. As a result, many ossifications have been dislocated, or are incompletely preserved and/or are partially preserved as imprints on the matrix. The outlines of many bones could only be partially or conjecturally deduced.

*Braincase.* The ventral and dorsolateral portions of the braincase and skull roof are exposed. We note that the skull roof does not seem to form a supraoccipital crest. The frontal bone seems to be the largest ossification of the cranial roof and forms a medial extension into the parietals. It extends above the whole length of the orbit without forming strong orbital notches. The skull roof is unornamented. A stocky, subquadrangular bone that forms a ventral process

is located at the anterodorsal corner of the orbit. Due to its position and ventral process, it is better treated as a lateral ethmoid. The putative lateral ethmoid is overlapped anteroventrally by a smooth bone fragment, which might correspond to the ventral expansion of the (dermo) mesethmoid. The ventral portion of the braincase and associated dermal ossifications are exposed. We can confidently report the presence of a lunate vomerine toothplate, which bears tiny tooth alveoli. The parasphenoid is straight throughout its length. The anterior portion of the parasphenoid, which lies below the orbits is wide, but the bone becomes narrower posteriorly, at the conceived level of the basisphenoid.

*Jaws.* The left premaxilla is preserved as a detailed imprint. It bears a stocky and slightly posteriorly inclined symphyseal process. The ascending and articular processes are weakly separated. There is a short and broad postmaxillary process, which lacks a gadoid notch. The alveolar process bears traces of tiny tooth alveoli along its oral length. The left maxilla is partially obstructed by the vomer and the left palate. The main body of the maxilla is straight, slightly taller posteriorly, edentulous and was largely excluded from the jaw gape. The supramaxillae are not preserved or were absent in vivo. Detailed imprints and fine sheaths of bone belonging to both lower jaws are preserved. The dentary is the largest bone of the lower jaw. The near symphyseal portion of the bone is narrow and medially to medioventrally curved. Posteriorly the dentary gains height to reach the height of the coronoid process of the anguloarticular. The posterior margin of the dentary, which receives the anguloarticular is deeply indented, forming a deep V. The ventral, postsymphyseal margin of the dentary is plate-like. The coronoid process of the anguloarticular is well-developed and rounded. The articulation surface of the anguloarticular is wide and faces dorsally. A faint trace of a bone near the posteroventral corner (upside down) of the right anguloarticular is tentatively attributed as a disarticulated retroarticular. Specimen AMPG_VTS_5 (Fig 5C) represents an isolated dentary that fits the description above. In addition, its medial surface exhibits an anteriorly widened alveolar surface, suggesting that multiple bands of teeth were present.

*Palate and suspensorium.* The fan-shaped quadrate is the only bone the outline of which can be directly observed. The anterior margin is almost straight to weakly sigmoidal, while the posterior margin is convex. The dorsal margin of the quadrate is continuous and strongly convex. The articular condyle is directed anteroventrally. The remaining ossifications are somewhat difficult to interpret, as they are mostly preserved as imprints. The precise boundaries of the otherwise massive metapterygoid could not be reliably reconstructed. Three to four edentulous, plate-like ossifications are situated anterodorsally to the quadrate, anteriorly to the metapterygoid and below the orbit. The dorsal-most bone is longer than tall, forms a dorsal concavity, and is best interpreted as an endopterygoid. Immediately ventrally to the putative endopterygoid, there is a partially-preserved ossification, which forms a dorsal convexity, and likely corresponds to the ectopterygoid. A trace of a shard-like dermopalatine is inferred to be wedged between the anterior tips of the putative ectopterygoid and endopterygoid. The ventral margin of the putative ectopterygoid is overlain by what appears to be either a cleanly broken or bipartite ossification, which in turn forms a lunate depression along its dorsal margin. If the breakage stands true then this ossification is best identified as an ectopterygoid, otherwise if this is accepted as a bipartite element then it most likely corresponds to a dislocated anterior and posterior ceratohyal. The boundaries between the possibly striated metapterygoid and the hyomandibula could not be traced. The hyomandibula appears to have been upright, forming two poorly differentiated dorsal articulation heads and, posteriorly, a stocky opercular process.

*Remaining circumorbital, cheek, opercular and pectoral girdle ossifications.* The preopercle forms a rather thin and long dorsal limb and a short, stocky anteroventral limb, which in turn forms a ventral projection. A thin ridge extends along the dorsal and ventral limbs of the preopercle. Remains of a plate-like bone located immediately anterior to the ventral limb of the

preopercle likely derive from the interopercle. Fragments corresponding to the opercle indicate the presence of a horizontal bar issuing from the articulation point of the bone. A thin shard of bone extends below and along the posterior length of the supposed preopercle, likely corresponding to the cleithrum. Traces of the pectoral fin, preceded by a postcleithral scale, can be seen at the ventral portion of the supposed cleithrum.

**Remarks.** AMPG_VTS_31 exhibits conspicuous differences from other taxa found in the Maastrichtian of Gavdos, yet its taxonomic and phylogenetic status cannot be accurately deciphered due to the incomplete preservation of many diagnostic structures. The jaws of AMPG_VTS_31 constitute its most diagnostic elements and allow us to constrain our attribution to Eurypterygii (sensu [45, 61]), on the basis of the following combination of characters (not necessarily synapomorphic): i) Presence of a clearly demarcated anterior process of the premaxilla; ii) presence of a postmaxillary process; iii) premaxilla being the principal dentigerous bone of the upper jaw, with maxilla being largely excluded from the jaw gape (for more information on those characters see [61, 88–92]). Several characters, such as, the absence of a well-developed occipital crest; the absence of a 'gadoid' notch on the postmaxillary process; the elongate and slender alveolar process of the premaxilla with regards to the anterior process; the poor differentiation between ascending and articular processes of the premaxilla, and the ventral position of the pectoral fin, allow us to constrain the taxonomic attribution of AMPG_VTS_31 to the earlier diverging eurypterygian clades Aulopiformes and Myctophiformes. These characters—or combinations of these—are typically not seen in members of derived eurypterygian lineages, such as various acanthomorphs and fossil allies [61, 89, 92–94]. Although the morphology of the anterior process of the premaxilla varies in aulopiforms [61, 66, 67], a postmaxillary process is typically absent in most fossil members of the clade, such as †*Nematonotus* [61], but it is present in some extant aulopiforms, like *Chlorophthalmus* [92]. Consequently, this loss is best treated as secondary.

Despite the larger size and overall robustness of the Greek specimen, its wide skull geometry; its premaxillary (robust symphyseal process; low postmaxillary process) and lower jaw anatomy (dentary shorter than anguloarticular and bearing an expanded ventral plate with an open sensory groove), as well as the presence of a large metapterygoid are reminiscent of the putative stem myctophiform †*Sardinioides frigoae* from the Late Cretaceous of Nardò, Italy [95]. Differences with the latter taxon can be found in the markedly convex (vs. straight) posterior margin of the quadrate, and possibly in the presence of a wider dorsal articulation surface of the hyomandibula in the Gavdian specimen. The Gavdian specimen bears a horizontal crest or strengthening bar on the opercle, which is absent in the better preserved †*Sardinioides monasteri* from the Campanian of Westphalia, Germany ([61, 78, 96]; T.A. pers. obs. on BSPG_AS_VII_877). The Westphalian species and especially its jaws are in all regards slenderer than those of the Gavdian specimen and †*Sardinioides frigoae*. Due to gestalt-based similarities with the latter, we opt for a tentative-only referral of the Gavdian material to †Sardinioididae. Its incomplete preservation precludes the study of some of the more diagnostic features for the group (e.g., midline contact of parietals; supraorbitals; number of supramaxillae; maxilla; shape of dermopalatine bone; see [66, 95, 96]).

Eurypterygii indet.

Fig 5D and 5E

**Material.** AMPG_VTS_16, left and right-side upper jaw bones

**Description.** The left and right premaxillae are preserved in articulation with their respective maxillae. The premaxillae are slender and elongate and each forms a stout, rounded dorsally to anterodorsally expanding anterior process. This anterior process is formed by an ascending process and a somewhat shorter and rounded, posteriorly bulging articular process. The two processes are weakly separated. The base of the anterior process is constricted. The

alveolar process is long and skinny and bears a short and elongate postmaxillary process that occupies the dorsal, mid-third of the former. The postmaxillary process does not form a posterior ('gadoid') notch. Teeth are not preserved, but must have been villiform, based on their corresponding tiny alveoli. The maxillae are also slender and elongate, but their posterior tip might be missing. They form a dorsomedially expanding articular process which is immediately followed posteriorly by a gentle dorsoventral constriction of the bone. The maxillae are edentulous and seem to have been excluded from the jaw gape in vivo. Their articular head seems to wrap around the articular portion of the anterior process of the premaxillae, and is immediately succeeded posteriorly by a shallow maxillary saddle (sensu [91]).

**Remarks.** These upper jaw bones and corresponding teeth, differ from those described, or expected, in most taxa from the Maastrichtian of Gavdos. They best resemble the jaws of AMPG_VTS_31, but the jaws of the latter are more robust. We therefore hypothesize that they possibly derive from a closely related but separate aulopiform or myctophiform taxon.

Teleostei indet.

Fig 5F

**Material.** AMPG_VTS_9, scale; AMPG_VTS_18, scale; AMPG_VTS_24, scale.

**Descripion.** These scales, often fragmentary, were common in the Vatsiana quarry debris. They are deeper than long and exhibit smooth outer surfaces comprising numerous closely packed and fine circuli.

**Remarks.** These scales better resemble those of Cretaceous †ichthyodectoids [55], but are best left unattributed until found in association with more diagnostic anatomical remains.

Chondrichthyes Huxley, 1880 [97]

Neoselachii Compagno, 1977 [98]

Hexanchiformes de Buen, 1926 [99]

Hexanchidae Gray, 1851 [100]

†*Gladioserratus* Underwood et al., 2011 [101]

†*Gladioserratus* sp.

Fig 5G

**Material.** AMPG_VTS_1, isolated lower? anterior tooth embedded in matrix.

**Description.** Complete tooth, labio-lingually compressed and mesio-distally elongate, with 13.4 mm in length and 9.1 mm in height; the specimen is visible only in labial view. The crown is smooth, characterized by a well-developed and triangular acrocone and three distal triangular cusplets, all of them smooth with sharp cutting edges, decreasing regularly in size distally. The mesial edge of the acrocone is inclined at an angle of approximately 33° in relation with the root base, and the heel of the edge has six small serrations, which are bent distally. The root is compressed with a flat labial face and an angular outline, and its mesial edge is relatively vertical and characterized by a concavity in its upper-middle part.

**Remarks.** The genus †*Gladioserratus* is represented by three recognized species: †*G. aptiensis* [102], †*G. magnus* [101], and †*G. dentatus* [103]. Underwood et al. [101] erected †*Gladioserratus* to differentiate Mesozoic species (†*G. aptiensis* and †*G. magnus*) from Cenozoic and recent *Notorynchus* species. †*Gladioserratus* can be distinguished from *Notorynchus* by its lower and mesially "rounded" root and by having cusps that are more massive and an evenly mesial serration [101, 104].

The morphological characters present in the crown and root of AMPG_VTS_1 from the Maastrichtian of Gavdos Island, match with the diagnosis of the genus †*Gladioserratus* [101]. Crown and root morphology in AMPG_VTS_1 also resemble lower? antero-lateral teeth of †*G. aptiensis* from the Aptian of France (see [105]: pl. II, Figs 3–7; [106]: pl. 2, Figs 6–10), and Cenomanian of Germany ([107]: Fig 2). Although certain differences in the morphology and inclination of the acrocone among AMPG_VTS_1 and the Aptian and Cenomanian specimens

could be noticed. Certain similarities also exist between AMPG_VTS_1 and the presumed first lower tooth †*G. magnus* from early Cenomanian of India ([101]: Fig 3C and 3D); although in the latter, the mesial edge outline of the root is relatively straight. In reference to †*G. dentatus* from the Valanginian of France ([103]: Fig 3G–3R), clear differences concerning the shape and inclination of the acrocone, distal cusplets, serrations of the mesial edge and root morphology allow us to differentiate it from AMPG_VTS_1. Few specimens from the Danian (Paleocene) of Denmark and Sweden have been recognized as †*Gladioserratus* sp. [104, 108]. These Scandinavian specimens also resemble the specimen from Gavdos. Underwood et al. [101] suggested that the Scandinavian teeth lie somewhere between †*Gladioserratus* and *Notorynchus*, as the root is much like the former but the crown is more like the latter with its cockscomb-like serration. Due to the absence of more material from Gavdos, AMPG_VTS_1 cannot be referred to a species. Despite its patchy fossil record, the stratigraphic range of †*Gladioserratus* spans the Aptian–Paleocene interval [101, 103, 107], although Adolfssen and Ward [104] suggested that this range could be extended to the early Eocene, if †*Notorynchus seratissimus* is included in the genus. The presence †*Gladioserratus* sp. in Gavdos represents the first fossil record of the genus from the Maastrichtian Stage, bridging a gap in the fossil record of the genus. This occurrence, which is the first one in Greece, increases the scarce knowledge of chondrichthyans from the region.

## Discussion

Prior to this work, only two 'fish' taxa were known from the mid–late Maastrichtian of Gavdos, a possible †saurodontid †ichthyodectoid and †*Enchodus* cf. *dirus* [21]. The newly collected and described material contains a maximum of nine taxa, which show little anatomical overlap with morphotypes previously known from the island. Despite the fact that we cannot attribute all morphotypes to low taxonomic levels (family or below), a conservative combined approach of the constituents of the assemblage implies the presence of a minimum of eight actinopterygian taxa belonging to †Ichthyodectoidei (two spp.), †Dercetidae (one sp.), †Enchodontidae (three spp.), †Ichthyotringidae (one sp.) and possibly †Sardinioididae (one or two sp.), and the hexanchid shark †*Gladioserratus* sp. The latter approach assumes that: i) at least one of the two unidentified †enchodontid morphotypes described in this work belongs to †*Enchodus* cf. *dirus* [21], ii) the unidentified eurypterygian is merged with the putative †sardinioidid and iii) the isolated cycloid scales derive from the two †ichthyodectoids. We note that the two †ichthyodectoid morphotypes recognized in the Maastrichtian of Gavdos are best treated as separate taxa, since the newly described fossil exhibits a tail configuration that best resembles that of the †cladocyclid †*Eubiodectes*, rather than that of †saurodontids (see also [55] and references therein). The lithographic preservation of the Gavdian assemblage, allows for the preservation of semi-articulated remains referrable to new species, such as †*Ichthyotringa pindica* and †*Calypsoichthys pavlakiensis*, which lend further support to previous claims of heightened aulopiform diversity immediately prior to the K–Pg Extinction [1, 87]. Prior to its discovery in the Maastrichtian of the Pindos Unit ([11]; this work) the genus †*Ichthyotringa* was thought to only reach the Campanian [96]. At the same time, the stratigraphic range of †Sardinioididae—if correctly identified as such—can be tentatively drawn from the early Campanian [109] horizons of the paleogeographically neighboring site of Nardò [95] into the mid-late Maastrichtian. An even greater range extension might hold true for the putative †cladocyclid [55]. Lastly, the single tooth of †*Gladioserratus* sp. from Gavdos constitutes the only known Maastrichtian occurrence of this possible K–Pg survivor [101, 104]. In total, the now better-known assemblage from Gavdos offers a rare glimpse into the offshore and deep-water ichthyofaunas from

the Maastrichtian of the Tethys, thus, helping to ameliorate a major deficiency of the global K–Pg fossil record (see also [1, 3, 11]).

## Paleoenvironmental implications of the assemblage

Previous lithostratigraphic and micropaleontological studies ascribed the depositional environment of the fossiliferous marly limestone layers of Pindos Unit in Gavdos to the offshore/bathyal realm [21, 28, 29] and implied the possible presence of anoxic or dysoxic bottom conditions [21]. Although most teleosts represented in our sample lack any close extant relatives (e.g., at the level of family or below), which can be more securely employed as ecological analogues, the overall composition of the assemblage seems to agree with an offshore and deep-water character. Specifically, we note the presence of multiple groups commonly found in offshore deposits and which are associated with epipelagic predatory lifestyles, such as †ichthyodectoids and †enchodontids [3, 21, 55, 110]. At the same time, an overview of the fossil record of †ichthyotringids and †sardinioidids (but see [95]) points towards deep and/or open water preferences for these taxa (see e.g., deep-water assemblages in Lebanon [83]; Morocco [74, 111]; Germany [96]; Eurytania [11]). The shearing dentition of †*Calypsoichthys* is hereby interpreted as a conceivable adaptation towards the consumption of soft-bodied prey like coleoids, as also implied by similar dentition in the jaws of recent meso–bathypelagic alepisaurid aulopiforms [112, 113]. We therefore tentatively hypothesize that †*Calypsoichthys* occupied a similar niche. Unlike in Eurytania, continental Greece [114], we have not yet uncovered fossils of soft-bodied cephalopods from the fossiliferous horizons examined here. The only macroinvertebrate fossil found at the site corresponds to a pair of ammonoid aptychi (AMPG_VTS_19). If modern hexanchids can be taken as an analogue [112], †*Gladioserratus* was likely a bathydemersal organism. This is further supported by the fossil record of the genus (see [103, 104]). Despite previous interpretations of oxygen depletion near or at the bottom [21] and taking into account that the incomplete preservation might at least partially reflect post-mortem transportation, the studied assemblage records taxa that normally inhabited different niches in the water column, ranging from epipelagic to bathydemersal.

## Comparison with other Maastrichtian sites from the 'Mediterranean' portion of the Tethys

The now better-known actinopterygian components of the Gavdian assemblage allow us to attempt some comparisons with previously described penecontemporaneous assemblages in continental Greece, which are lithostratigraphically placed in the same paleogeographic realm (Pindos Unit), as well as with other Maastrichtian assemblages from the Tethys. The discussion is largely restricted to actinopterygians from said assemblages, since the Gavdian assemblage so far contains a single chondrichthyan tooth. This sample incomparable to the extensive ones available from other Maastrichtian sites of the Tethys and neighboring sedimentary basins (e.g., Spain [115, 116], North Africa: [15, 117, 118], Niger [119], the Middle East [14, 120, 121] and India [122, 123]).

The mid–late Maastrichtian assemblage of Gavdos is characterized by a similar type of lithographic preservation as the recently rediscovered late Maastrichtian localities in Eurytania, continental Greece [11, 64], and in some regards the two assemblages seem to complement each other (Table 1). With few exceptions, the actinopterygian material collected from both regions comprises semi-articulated or disarticulated individuals of various size ranges, which cannot always be taxonomically attributed to low taxonomic levels. Fossils are seemingly more abundant in Eurytania (T.A. pers. obs.), while the confirmed vertebrate biodiversity from the area includes up to ten actinopterygian and at least three shark taxa ([11, 64]; T.A. and J.D.C.

**Table 1. Updated faunal lists for the Maastrichtian of the Pindos Unit in Gavdos and Eurytania.**

| Higher taxa | Gavdos Island | Eurytania |
| --- | --- | --- |
| **Teleostei** | | |
| **†Ichthyodectiformes** | †Ichthyodectoidei indet. 1 (?†saurodontidae) [21] | |
| | †Ichthyodectoidei indet. 2 * | |
| Elopiformes | | Elopomorpha indet. [11, 64] |
| **Aulopiformes** | | |
| †Dercetidae | †Dercetidae indet.* | †Dercetidae indet. [11, 64] |
| †Enchodontidae | †*Enchodus* cf. *dirus* [21] | †Enchodontidae indet. 3 [11] |
| | †Enchodontidae indet.1* | †Enchodontidae indet. 4 [11] |
| | †Enchodontidae indet.2* | |
| | †*Calypsoichthys pavlakiensis** | †*Calypsoichthys* sp.* [11] |
| †Ichthyotringidae | †*Ichthyotringa pindica** | †*Ichthyotringa pindica** [11] |
| **Eurypterygii incertae sedis** | Eurypterygii indet. 1* | Eurypterygii indet. 2 [11] |
| †Sardinioididae | cf. †Sardinioididae indet.* | |
| **Teleostei incertae sedis** | Teleostei indet.1 | Teleostei indet. 2 [11] |
| | | Teleostei indet. 3 [11] |
| | | Teleostei indet. 4 [11] |
| **Neoselachii** | | |
| **Hexanchiformes** | | |
| Chlamydoselachidae | | Chlamydoselachidae indet. [64] |
| Hexanchidae | †*Gladioserratus* sp.* | Hexanichidae indet. [64] |
| **Lamniformes** | | Lamniformes indet. [11] |

This table lists all discrete morphotypes identified as separate taxa from the Maastrichtian of the Pindos Unit. References are given in superscript, following each taxon. Eurytanian taxa listed here are limited only to those figured by Koch and Nicolaus [64] and/or confirmed or discovered by Argyriou and Davesne [11]. The numbering of morphotypes left in open nomenclature has been changed from [11], for clarity. The asterisk (*) indicates taxa first described in this work.

pers. obs). Unsurprisingly, the position of the Maastrichtian fossiliferous horizons in Eurytania and Gavdos within the same pelagic paleogeographic realm of the Pindos Unit is also reflected in the shared presence of †*Ichthyotringa pindica*, the genus †*Calypsoichthys* and what appears to be the same †dercetid morphotype. To date, the former two taxa are only known from the Pindos Unit. On the contrary, possible elopomorphs, a small-sized unidentified eurypterygian with long pectoral fins and a teleost with pointed and serrated teeth, which are present in Eurytania [11, 64], have not yet been recognized in Gavdos. As of now, no fossils reliably attributed to †ichthyodectoids or †sardinioidids have been described from continental Greece. A more diverse assemblage of chondrichthyans was reported from Eurytania, with the genera *Hexanchus*, *Isurus*, "†*Corax*", *Odontaspis* and *Squalus* reported—most of which were inaccurately ascribed to Cenozoic genera and never figured or described—by Koch and Nicolaus [64]. The only specimens illustrated by Koch and Nicolaus ([64]: taf 33, Figs 4 and 5) include an isolated hexanchid tooth reported as †*Notidanus microdon* (= †*Hexanchus microdon*; see [124]), and an indeterminate chlamydoselachid tooth, both of which are indicative of deep-water environments [112]. A further indeterminate large lamniform tooth was recently found by the first author (T.A. pers. obs.), raising the number of confirmed shark taxa to three. Given the differences in the age of the first terrigenous (flysch) deposits between Eurytania (latest Maastrichtian–Danian, sensu [64, 125] and Gavdos (early–middle Eocene, sensu [28, 29]), it can be hypothesized that Gavdos was located further from the advancing land than continental Greece during the Maastrichtian, and that its contact with deeper neotethyan oceanic basins

was better established. This might have influenced its faunal composition, but further sampling in both regions is required to evaluate such differences.

The majority of Maastrichtian actinopterygian body fossils from the Mediterranean portion of the Tethys comes from phosphatic horizons, which were deposited in rather shallow epicontinental platform environments along its 'Gondwanan shore'. Most known Tethyan assemblages are located in what is nowadays Egypt [126, 127], Saudi Arabia [16], Israel [17, 19], Jordan [128, 129] and Syria [14]. Extensively commercially exploited and studied fossiliferous deposits of similar nature to those of the Middle East are also known from Morocco, but they were most-likely connected to the Atlantic [13, 15]. These often winnowed assemblages [1, 17] date to the early and middle Maastrichtian and typically favor the preservation of ichthyolites (isolated bones and teeth) mid–large-sized taxa, and are dominated by the epipelagic †*Enchodus* spp. and †*Stratodus*? *apicalis*, or neritic †pycnodontids. Aside from those of the Pindos Unit ([11, 21]; this work), a single fossiliferous locality is so far known from what is now the northern coast of the Mediterranean; the inconclusively dated but possibly early–mid Maastrichtian shallow marine or brackish assemblage of Trebiciano [130–132], in Italy. We note that the minimum age estimate for the Italian 'Melissano Limestone', which hosts the fossil-rich neritic vertebrate assemblage of Nardò (>40 recognized 'fish' spp., see faunal list in [133]), was pushed back from the Campanian–early Maastrichtian [134] to the early Campanian [109]. Thus, the Nardò fossils should better be excluded from discussions on the Maastrichtian vertebrate diversity of the Tethys (see also [3]). When other Tethyan Maastrichtian assemblages are compared with e.g., those of the fossil-rich type Maastrichtian strata (16 spp.: [12, 135, 136]), the recognized actinopterygian diversity in Tethyan sites is rather low, ranging between seven (Middle Eastern Phosphates [14, 17, 19]), to approximately ten distinct morphotypes (Trebiciano [130–132]).

With a minimum of eight distinct actinopterygian morphotypes, the Gavdian assemblage abides to the commonly inferred motif of Tethyan actinopterygian oligotypism during the Maastrichtian, although this might be attributed to the small sample size available from the site. A slightly different picture emerges when the all actinopterygian morphotypes confirmed to be present in the Maastrichtian of the Pindos Unit are treated collectively [11] (Table 1). This way, and after lumping the unidentified †enchodontid taxa, as well as some of the unidentified teleosts together, the recognized diversity of the Greek Maastrichtian deposits climbs to a minimum of 14 discrete morphotypes. Most of these are still known from disarticulated material. To the exception of †Ichthyotringidae and the possible †sardinioidid, the higher taxa (†Ichthyodectoidei; †Enchodontidae) represented in Gavdos are known from the Maastrichtian sites of North Africa and the Middle East, while there is no taxonomic overlap with Trebiciano. The fossil record of †Enchodontidae from the Maastrichtian was previously restricted to numerous occurrences of various species more or less securely ascribed to †*Enchodus* on the basis of isolated teeth and dentigerous bones [12–15, 18, 19, 21], as well as dubious occurrences of unspecified †eurypholins [64]. †*Calypsoichthys* exhibits dermopalatine bones with a single fang that, if found disarticulated, could be easily confused with those of †*Enchodus*. Its recognition as a distinct genus is indicative of the presence of multiple sub-lineages of †Enchodontidae in the Maastrichtian of the Tethys and beyond, which possibly proliferated in different ecological niches and paleoenvironments. The same probably applies to †ichthyotringids, the putative †sardinioidid, and †*Gladioserratus*, which were previously unknown from the Maastrichtian. It thus becomes evident that one of the reasons that the Maastrichtian 'fish' biodiversity has been historically underappreciated is the scarcity of fossiliferous deposits from calm, deep-water paleoenvironments.

## Conclusions

The newly collected material from the mid–late Maastrichtian of Pindos Unit in Gavdos Island reveals the presence at least seven actinopterygian morphotypes previously undescribed from the site, as well as a hexanchid shark. Although the studied fossil sample is small and contains mostly partially preserved remains, it allows for the first description of the †enchodontid †*Calypsoichthys pavlakiensis* and the †ichthyotringid †*Ichthyotringa pindica*. The moderately diverse assemblage is indicative of deep and open water depositional environments, which are otherwise poorly represented in the known body-fossil record of the latest Cretaceous (see discussion in [11]). It is noteworthy that many recent works have highlighted the important role that such undersampled Cretaceous–Paleocene pelagic and deep-water environments might have played in the evolution of various neoteleostean clades, such as aulopiforms, myctophids, gadoids and various percomorphs [10, 113, 137]. The Gavdian assemblage consolidates the proposed extension of the stratigraphic range of †Ichthyotringids [11] and †*Gladioserratus* into the latest Cretaceous, while the same might hold true for †sardinioidids. We anticipate that additional vertebrate diversity is hidden in the deep-water facies of the Pindos Unit in Greece, and we stress the importance of the continuation of research on these sites, which can lead to a better understanding of the K–Pg Extinction turnover of deep-water Tethyan ichthyofaunas. As a note for future investigations, the marly-limestone sedimentation in Gavdos continues into the late Maastrichtian and the Paleocene, and possibly records the K–Pg boundary [28, 29]. These younger horizons can be accessed in the area near the lighthouse [28], but they have not yet been sampled for vertebrates.

## Acknowledgments

We are indebted to the mayor, Ms. L. Stefanakis, and deputy mayor, Mr. S. Maravelakis, of Gavdos Island, for facilitating our collection efforts. Precious administrative support, fieldwork permits and access to the infrastructure and collections of the AMPG were kindly provided by S. Rousiakis, E. Koskeridou and V. Karakitsios (all NKUA and AMPG). We extend our Acknowledgments towards O. Otero (Univ. Poitiers) and G. Clément (MNHN), as well as the members of the FOSFO workgroup of the CR2P, MNHN, for greatly facilitating the research stay of TA at the MNHN. O. Otero is also thanked for facilitating access to recent comparative material of teleosts. A. Lòpez-Arbarello (LMU) and D. Davesne (MFN) are thanked for useful discussions on anatomy, and R. Kindlimann for generously offering access to comparative material of elasmobranchs. Thanks to M. Sánchez-Villagra and the Evolutionary Morphology and Palaeobiology group at the Palaeontological Institute and Museum of the University of Zurich for supporting the work of JDCB. Finally, we would like to thank P.M. Brito (State Univ. Rio de Janeiro) and an anonymous reviewer, as well as editor G. Carnevale (Univ. Torino) for their reviews.

## Author Contributions

**Conceptualization:** Thodoris Argyriou, Lionel Cavin.

**Data curation:** Thodoris Argyriou.

**Formal analysis:** Thodoris Argyriou, Jorge D. Carrillo-Briceño, Lionel Cavin.

**Funding acquisition:** Thodoris Argyriou.

**Investigation:** Thodoris Argyriou, Apostolos Alexopoulos, Jorge D. Carrillo-Briceño, Lionel Cavin.

**Project administration:** Thodoris Argyriou.

**Resources:** Apostolos Alexopoulos.

**Writing – original draft:** Thodoris Argyriou, Apostolos Alexopoulos, Jorge D. Carrillo-Briceño, Lionel Cavin.

**Writing – review & editing:** Thodoris Argyriou, Apostolos Alexopoulos, Jorge D. Carrillo-Briceño, Lionel Cavin.

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
