## [Decision Letter · Decision Letter 0]

21 Jan 2022

PONE-D-21-41103A fossil assemblage from the mid–late Maastrichtian of Gavdos Island, Greece, provides insights into the pre-extinction pelagic ichthyofaunas of the Tethys.PLOS ONE

Dear Dr. Argyriou,

Thank you for submitting your manuscript to PLOS ONE. After careful consideration, we feel that it has merit but does not fully meet PLOS ONE’s publication criteria as it currently stands. Therefore, we invite you to submit a revised version of the manuscript that addresses the points raised during the review process.

We look forward to receiving your revised manuscript.

Kind regards,

Giorgio Carnevale, Ph.D

Academic Editor

PLOS ONE

Journal Requirements:

2. For sentences please see Template Letters Appendix of the New Species protocol in the wiki – New Species Protocol and email sentences

4. We note that Figures 2 to 5 in your submission contain copyrighted images. All PLOS content is published under the Creative Commons Attribution License (CC BY 4.0), which means that the manuscript, images, and Supporting Information files will be freely available online, and any third party is permitted to access, download, copy, distribute, and use these materials in any way, even commercially, with proper attribution. For more information, see our copyright guidelines: http://journals.plos.org/plosone/s/licenses-and-copyright.

a. You may seek permission from the original copyright holder of Figures 2 to 5 to publish the content specifically under the CC BY 4.0 license. 

Reviewers' comments:

Reviewer's Responses to Questions

**Comments to the Author**

1. Is the manuscript technically sound, and do the data support the conclusions?

Reviewer #1: Yes

Reviewer #2: Yes

2. Has the statistical analysis been performed appropriately and rigorously? 

Reviewer #1: N/A

Reviewer #2: N/A

3. Have the authors made all data underlying the findings in their manuscript fully available?

Reviewer #1: Yes

Reviewer #2: Yes

4. Is the manuscript presented in an intelligible fashion and written in standard English?

Reviewer #1: Yes

Reviewer #2: Yes

5. Review Comments to the Author

Reviewer #1: I believe that this work offers new information about a fossil association hitherto poorly understood. I recommend that this work be published, but first, the authors should consider the following observations.

I think the authors should reconsider the determination of its ichthyodectiforme, I think it will be more correct to point it out as an indeterminate member of the suborder Ichthyodectoidei. They should also correct this information in the Table 1.

In the attached pdf mark in yellow some writing errors that must be corrected before accepting your work for publication.

Reviewer #2: Dear authors,

I consider your manuscript an important contribution to the knowledge of the ichthyofauna that predates the K/T extinction. In addition to expanding the list of taxa in the Pinos Unit, confirming previous interpretations of an offshore and very deep depositional environment for this unit, you describe two new taxa, which further increase our knowledge of the Maastrichtian fauna.

The article is very clear, well-formulated, with an up-to-date bibliography and good figures.

For this reason, I can only congratulate the authors for their excellent work.

6. PLOS authors have the option to publish the peer review history of their article (what does this mean?). If published, this will include your full peer review and any attached files.

Reviewer #1: No

Reviewer #2: **Yes: **Paulo M. Brito

---

## [Author Response · Author response to Decision Letter 0]

5 Feb 2022

We are indebted to the reviewers and editor for their careful and positive evaluation of our paper. Below we expand on the main comments made by reviewer 1 in their review letter and on the marked pdf, as well as on the comments made by the editorial stuff of the journal.

Reviewer 1:

General comment on †Ichthyodectoidei: We appreciated and accepted the reviewer’s suggestion to narrow down the attribution of our †Ichthyodectiform indet. We have added the reference of †Ichthyodectoidei and corrected/updated all relevant passages in the text and table 1.

Abstract and main text: We chose to put fishes in quotation marks, in order to make it clear that we refer to them as a paraphyletic assemblage, which includes chondrichthyans and actinopterygians but excludes sarcopterygians. We can remove the quotation marks if the editor wishes us to do so.

Comparative material section: As requested by the reviewer, we listed the naming authorities of various taxa and provided references for them.

In addition to the above, we accepted all writing suggestions made by the reviewer in their annotated pdf.

Additional changes:

Fig 1. We changed the color of the Pindos Unit in Fig 1B to grey, do it is not confused with the Cretaceous-aged platy limestone on Fig1C. The caption was updated accordingly.

Fig4B. We added a label for the dermopterotic ossification in both figure and caption

---

## [Editor Report · Decision Letter 1]

8 Mar 2022

A fossil assemblage from the mid–late Maastrichtian of Gavdos Island, Greece, provides insights into the pre-extinction pelagic ichthyofaunas of the Tethys.

PONE-D-21-41103R1

Dear Dr. Argyriou,

We’re pleased to inform you that your manuscript has been judged scientifically suitable for publication and will be formally accepted for publication once it meets all outstanding technical requirements.

Kind regards,

Giorgio Carnevale, Ph.D

Academic Editor

PLOS ONE

Additional Editor Comments (optional):

All the issues raised by the reviewers have been addressed and the manuscript is currently suitable for publication.
---

## [Editor Report · Acceptance letter]

22 Mar 2022

PONE-D-21-41103R1 

A fossil assemblage from the mid–late Maastrichtian of Gavdos Island, Greece, provides insights into the pre-extinction pelagic ichthyofaunas of the Tethys. 

Dear Dr. Argyriou:

I'm pleased to inform you that your manuscript has been deemed suitable for publication in PLOS ONE. Congratulations! Your manuscript is now with our production department. 

Kind regards, 

on behalf of

Dr. Giorgio Carnevale 

Academic Editor

PLOS ONE